# Large Language Models can be Guided to Evade AI-Generated Text Detection

**Ning Lu**[*]                                                                                                    *nluab@cse.ust.hk*
*Guangdong Key Laboratory of Brain-Inspired Intelligent Computation*
*Department of Computer Science and Engineering, Southern University of Science and Technology*
*Department of Computer Science and Engineering, Hong Kong University of Science and Technology*

**Shengcai Liu**[*,†]                                                                                              *liusccc@gmail.com*
*Centre for Frontier AI Research (CFAR), Agency for Science, Technology and Research (A\*STAR)*

**Rui He**                                                                                              *her2018@mail.sustech.edu.cn*
*Guangdong Key Laboratory of Brain-Inspired Intelligent Computation*
*Department of Computer Science and Engineering, Southern University of Science and Technology*

**Yew-Soon Ong**                                                                                              *asysong@ntu.edu.sg*
*Centre for Frontier AI Research (CFAR), Agency for Science, Technology and Research (A\*STAR)*
*College of Computing and Data Science, Nanyang Technological University (NTU)*

**Qi Wang**                                                                                              *wangqi@sustech.edu.cn*
*Department of Computer Science and Engineering, Southern University of Science and Technology*

**Ke Tang**                                                                                              *tangk3@sustech.edu.cn*
*Guangdong Key Laboratory of Brain-Inspired Intelligent Computation*
*Department of Computer Science and Engineering, Southern University of Science and Technology*

[*]*Equal contribution.* [†]*Corresponding author.*

**Reviewed on OpenReview:** *https://openreview.net/forum?id=lLEOmWzUrr*

## Abstract

Large language models (LLMs) have shown remarkable performance in various tasks and have been extensively utilized by the public. However, the increasing concerns regarding the misuse of LLMs, such as plagiarism and spamming, have led to the development of multiple detectors, including fine-tuned classifiers and statistical methods. In this study, we equip LLMs with prompts, rather than relying on an external paraphraser, to evaluate the vulnerability of these detectors. We propose a novel **S**ubstitution-based **I**n-**C**ontext example **O**ptimization method (SICO) to automatically construct prompts for evading the detectors. SICO is cost-efficient as it requires only 40 human-written examples and a limited number of LLM inferences to generate a prompt. Moreover, once a task-specific prompt has been constructed, it can be universally used against a wide range of detectors. Extensive experiments across three real-world tasks demonstrate that SICO significantly outperforms the paraphraser baselines and enables GPT-3.5 to successfully evade six detectors, decreasing their AUC by 0.5 on average. Furthermore, a comprehensive human evaluation show that the SICO-generated text achieves human-level readability and task completion rates, while preserving high imperceptibility. Finally, we propose an ensemble approach to enhance the robustness of detectors against SICO attack. [1]

---

[1]The code is publicly available at `https://github.com/ColinLu50/Evade-GPT-Detector`.

# 1 Introduction

The rapid advancement of large language models (LLMs), such as GPT (Brown et al., 2020) and LLaMa (Touvron et al., 2023), has led to a largely-increased capacity for generating high-quality human-like text. However, there are also growing concerns surrounding the misuse of these models, including generating fake product reviews (Adelani et al., 2020; Lin et al., 2022) and misinformation (Lin et al., 2022), enabling academic dishonesty (Stokel-Walker, 2022), and producing misleading answers on websites (StackOverflow, 2023).

In response to these challenges, several methods for detecting AI-generated text have been proposed recently, ranging from fine-tuned classifiers (Guo et al., 2023; Solaiman et al., 2019), statistical methods (Mitchell et al., 2023), to watermarking (Kirchenbauer et al., 2023). There are also online detection services provided by companies such as GPTzero (Tian, 2023). However, the robustness of these detection methods has not been thoroughly evaluated. Recent studies (Krishna et al., 2023; Sadasivan et al., 2023) have shown the vulnerability of these detectors to the so-called *paraphrase attacks*, which adopt an external paraphraser to rewrite the text generated by LLMs to evade detectors.

In this work, rather than relying on an external paraphraser, we explore equipping LLMs with carefully constructed prompts to evade detectors. The intuition is that, given the remarkable capabilities of LLMs, appropriate prompts can guide these models to potentially achieve and even exceed the evasion performance level of smaller external paraphrasers. We propose **SICO**, a **S**ubstitution-based **I**n-**C**ontext example **O**ptimization method, to automatically construct such prompts based on human-generated examples. Specifically, SICO iteratively substitutes words and sentences within the in-context examples to provide more representative demonstrations for LLMs to generate text that cannot be detected, where the substitution procedure is directed by a proxy detector (see Figure 1 for an overview of SICO).

We assess the evasion performance of SICO across three real-world tasks that are susceptible to the misuse of LLMs, i.e., academic essay writing, open-ended question answering, and fake review generation. The results demonstrate that SICO consistently outperforms the paraphraser baselines, leading to a decrease in AUC by approximately 0.5 on average for six existing detectors. Additionally, a comprehensive human evaluation involving 600 examples shows that the SICO-generated text is comparable to, and in some cases even better than, human-written text in terms of readability and task completion rates. It also demonstrates that SICO reduces the probability of being recognized by humans. In addition to its strong evasion performance, SICO is also cost-efficient and easy to use. Unlike paraphraser-based methods that often require extensive computational resources – as evidenced by the fine-tuning of a 13B model on a large dataset (Krishna et al., 2023) – SICO only requires 40 human-generated examples and a limited number of LLM inferences (e.g., costing approximately 1 USD using the GPT-3.5 API). Besides, once a task-specific prompt has been constructed by SICO, it can be universally used against a wide range of detectors.

Considering the importance of detecting AI-generated text to avoid their misuse, the results presented in this work certainly reveal the vulnerability of the existing detectors. Besides, this work presents the first empirical evidence that LLMs can evade detectors through a prompt-guided approach. The strong evasion performance of SICO suggests that it can be used as a standard evaluation tool for any future AI-generated text detectors. Finally, we propose an ensemble approach to enhance the robustness of detectors against SICO attack. We hope that these findings can better facilitate the research concerning the responsible use of LLMs. To summarize, our main contributions are:

- We introduce SICO, a novel in-context example learning method, to automatically construct prompts that can guide LLMs to evade detectors.

- With low cost, SICO achieves strong performance in evading six existing detectors across three tasks, significantly outperforming the paraphraser baselines.

- A comprehensive human evaluation verifies that the SICO-generated text achieves human-level readability and task completion rates, while preserving high imperceptibility.

## 2 Related works

### 2.1 AI-generated text detection

In recent years, the research community has developed a wide range of detectors for AI-generated contents. In general, these detectors can be classified into three categories: training-based, statistical, and watermarking methods. Training-based methods treat the detection problem as a binary classification task, where neural networks are trained using AI-generated text and human-written text. Early studies utilized classifiers to identify fake reviews (Hovy, 2016) and fake news (Zellers et al., 2019). More recently, researchers have trained classifiers using text generated by LLMs, such as the GPT-3.5 detector (Guo et al., 2023) and GPT-2 detector (Solaiman et al., 2019). Statistical methods, on the other hand, focus on zero-shot detection without any additional training overhead. These methods seek to distinguish between human-written text and AI-generated text based on the statistical characteristics of text, such as the statistical irregularities in measures like entropy (Lavergne et al., 2008), perplexity (Beresneva, 2016) and token rank (Gehrmann et al., 2019). A recent method, DetectGPT (Mitchell et al., 2023), exploits the phenomenon that AI-generated text tends to lie in the negative curvature regions of log probability of text. The watermarking methods involve modifying the LLM's text generation process to imprint specific patterns on the generated text, such that it can be detected (Abdelnabi & Fritz, 2021; Grinbaum & Adomaitis, 2022; Kirchenbauer et al., 2023). Although the proposed method SICO primarily focuses on the first two types of detection methods, it can also help evade watermarking when acted as an external paraphraser, as shown in Appendix F.

Recent studies have found that paraphrasing can evade these detectors, which trains an additional neural network to rewrite the original AI-generated text (Krishna et al., 2023; Sadasivan et al., 2023). In contrast, SICO eliminates the need for extra models or training steps. SICO provides an automatic approach that iteratively improves prompt, unlike the in-the-wild prompts, which which are typically discovered through manual trial and error (Uploader, 2023).

### 2.2 In-context learning

With the increasing scales of models and corpora (Radford et al., 2019; Chowdhery et al., 2022; Gou et al., 2022; Wei et al., 2024), LLMs have demonstrated the in-context learning (ICL) ability, allowing them to perform tasks with only a few examples provided as demonstrations (Brown et al., 2020). Recent studies have focused on designing demonstrations during inference, which can be divided into demonstration selection, ordering, and formatting (Dong et al., 2022). Specifically, demonstrations can be selected based on unsupervised metrics or supervised strategies (Kim et al., 2022; Gonen et al., 2022; Wei et al., 2023). For ordering, Liu et al. (2021) sorted examples by their distances to the input. Regarding demonstration formatting, Wei et al. (2022) proposed the so-called chain-of-thoughts (COT) format, and subsequent works have developed automatic COT (Zhang et al., 2022). In contrast to these works, we focus on iteratively optimizing demonstrations through substitutions. In principle, the proposed method SICO can be used in combination with the above-mentioned methods, potentially leading to improved performance.

## 3 Substitution-based in-context example optimization (SICO)

The illustration of SICO is presented in Figure 1. First, LLM is asked to extract language features of human-written text. Then, the in-context examples are initialized and optimized. The final prompt is composed of the feature, task instruction, and optimized in-context examples. Below, we first describe how to evaluate a prompt during its optimization and then elaborate all the steps of SICO.

### 3.1 Prompt Evaluation

Given a natural language processing task, denote the task input as $x$. To assess the utility of a prompt $p$, we first collect a set of task inputs, $X_{eval}$. For each input $x \in X_{eval}$, $p$ and $x$ are first concatenated (denoted by $p \oplus x$) and fed into the LLM, whose output text (denoted by $\text{LLM}(p \oplus x)$) is then classified by a proxy detector. Let $\mathcal{P}_{\text{AI}}$ be the predicted probability of $\text{LLM}(p \oplus x)$ to be AI-generated, then the utility score of

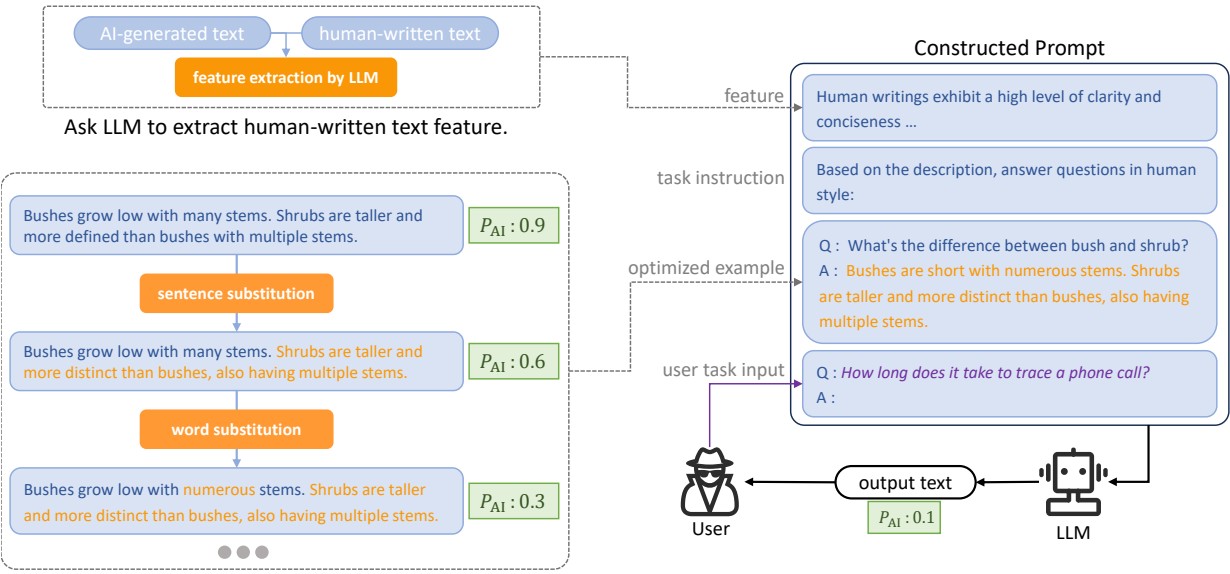

Figure 1: Illustration of how SICO generates prompts for the question answering task. The probability $P_{\mathrm{AI}}$, as predicted by the proxy detector, indicates the likelihood that the given text is AI-generated. Once SICO prompt is constructed, it serves as a template, allowing users to insert various task inputs (highlighted in purple text).

prompt $p$, denoted by $\mathcal{U}(p)$, is defined as one minus the averaged predicted probability across $X_{eval}$ (the higher $\mathcal{U}$, the better):

$$\mathcal{U}(p) = 1 - \frac{1}{|\mathbf{X}_{eval}|} \sum_{x \in X_{eval}} \mathcal{P}_{\mathrm{AI}}(\mathrm{LLM}(p \oplus x)). \tag{1}$$

### 3.2 Prompt Construction

**Data collection** We first collect a set of $K$ triplets, i.e., $D = \{(x_{\mathrm{ic}}^k, y_{\mathrm{AI}}^k, y_{\mathrm{human}}^k)\}_{k=1}^K$, where $x_{\mathrm{ic}}^k$ is a task input and $y_{\mathrm{AI}}^k, y_{\mathrm{human}}^k$ are the corresponding outputs generated by the LLM and humans, respectively. Note $D$ is used for prompt construction and it is independent of $X_{eval}$ which is used for prompt evaluation.

**Feature extraction** This step involves the $K$ pairs of AI-generated and human-written outputs from $D$, denoted by $\{(y_{\mathrm{AI}}^k, y_{\mathrm{human}}^k)\}_{k=1}^K$. We provide LLM with these pairs and ask LLM to extract the distinct linguistic features of human-written text, denoted as $t_{\mathrm{feature}}$. The top left of Figure 1 demonstrates this process and generate text to describe the feature of human-written text. This feature text is then utilized for sentence paraphrasing and included in the final prompt.

**In-context example optimization** We initialize the in-context examples as $(x_{\mathrm{ic}}^k, y_{\mathrm{ic}}^k)$, where $y_{\mathrm{ic}}^k$ is generated by paraphrasing $y_{\mathrm{AI}}^k$. More specifically, the feature $t_{\mathrm{feature}}$ is concatenated with a paraphrasing instruction to instruct LLM to paraphrase the AI-generated text to obtain the initial $y_{\mathrm{ic}}^k$. The paraphrasing instruction is "*Based on the description, paraphrase the following text to be human style*".

Then the in-context output $y_{\mathrm{ic}}$ is iteratively optimized to be less AI-like, as determined by the probability $\mathcal{P}_{\mathrm{AI}}$ calculated by the proxy detector. By presenting more and more representative, i.e. lower $\mathcal{P}_{\mathrm{AI}}$, in-context demonstrations to LLM, it is expected to understand how to generate human-like outputs. This in-context example optimization procedure is the key step in SICO for improving evasion performance, as verified by the ablation study in Section 5.1. Formally, the optimization goal can be expressed as:

$$y_{\mathrm{ic}}^* = \underset{y_{\mathrm{ic}}' \in \mathrm{SIM}(y_{\mathrm{ic}})}{\arg\min} \, \mathcal{P}_{\mathrm{AI}}(y_{\mathrm{ic}}'), \tag{2}$$

---

**Algorithm 1** Substitution-based in-context example optimization (SICO)

---

**Require:** large language model LLM, prompt utility function $\mathcal{U}(\cdot)$, $D = \{(x_{\mathrm{ic}}^k, y_{\mathrm{AI}}^k, y_{\mathrm{human}}^k)\}_{k=1}^K$, $\mathbf{X}_{eval}$, total iteration number $N$

1: Extract language feature $t_{\mathrm{feature}}$ using $\{(y_{\mathrm{AI}}^k, y_{\mathrm{human}}^k)\}_{k=1}^K$ and LLM
2: Construct in-context outputs $y_{\mathrm{ic}}^k = \mathrm{LLM}(t_{\mathrm{feature}} \oplus p_{\mathrm{para}} \oplus y_{\mathrm{AI}}^k), \ \forall k \in \{1, ..., K\}$
3: Initialize $p^* \leftarrow t_{\mathrm{feature}} \oplus p_{\mathrm{task}} \oplus \{(x_{\mathrm{ic}}^k, y_{\mathrm{ic}}^k)\}_{k=1}^K$
4: **for** $n = 1$ to $N$ **do**
5:    **for** $k = 1$ to $K$ **do**
6:       Generate sentence-level / word-level substitutions $C^k$ of $y_{\mathrm{ic}}^k$, switching based on $n$
7:       Optimize $y_{\mathrm{ic}}^k$ using Algorithm 2: $\hat{y}_{\mathrm{ic}}^k \leftarrow \mathrm{GreedyOPT}(y_{\mathrm{ic}}^k, C^k)$
8:    **end for**
9:    Construct new prompt $\hat{p}$: $\hat{p} \leftarrow t_{\mathrm{feature}} \oplus p_{\mathrm{task}} \oplus \{(x_{\mathrm{ic}}^k, \hat{y}_{\mathrm{ic}}^k)\}_{k=1}^K$
10:    **if** $\mathcal{U}(\hat{p}) > \mathcal{U}(p^*)$ **then**
11:       Update in-context examples $y_{\mathrm{ic}}^k \leftarrow \hat{y}_{\mathrm{ic}}^k$ and update the best prompt $p^* \leftarrow \hat{p}$
12:    **end if**
13: **end for**
14: **return** $p^*$

---

where $\mathrm{SIM}(y_{\mathrm{ic}})$ denotes the set of text that is semantically similar to $y_{\mathrm{ic}}$. The goal of setting such semantic restriction is to maintain the usability of the text during optimization. In SICO, we generate semantically similar text by replacing words and rephrasing sentences. This is explained in detail below.

**Substitution type** To generate $y_{\mathrm{ic}}'$ that is semantically similar to $y_{\mathrm{ic}}$, we employ substitution at word level and sentence level in turn. For word-level substitution, we use WordNet (Miller, 1998), a lexical database of English words, to construct a synonym substitution set. We restrict substitutions to content words[2] and ensure that the substitution would not change the part-of-speech tags. We use a mask language model to filter out the candidate words that not fits the context. For sentence-level substitution, we utilize the paraphrasing instruction combined with extracted feature, denoted as $t_{\mathrm{feature}} \oplus p_{\mathrm{para}}$. This combined instruction is used to prompt LLM to generate paraphrases for each $y_{\mathrm{ic}}$. Using $t_{\mathrm{feature}}$ will make the generated paraphrases be more human-like, thus increasing the efficiency of optimization.

**Algorithm** As illustrated in Algorithm 1, SICO would optimize $\{y_{\mathrm{ic}}^k\}_{k=1}^K$ for $N$ iterations (lines 4-17). At each iteration, each $y_{\mathrm{ic}}^k$ would be optimized by greedy substitution (line 11), as presented in Algorithm 2. Specifically, for the $i$-th original word/sentence $y_i$ in the text $y$, let $C_{i,j}$ denote its $j$-th synonym/paraphrase, and let $\mathrm{SUB}(y_i, C_{i,j})$ denote the new text resulting from substituting $y_i$ with $C_{i,j}$. For each substitution position $i$, SICO identifies the best substitution $C_{i,*}$ by checking which $C_{i,j}$ results in the lowest AI probability when replacing $y_i$ (Line 1 in Algorithm 2).

After obtaining the optimized in-context output $\hat{y}_{\mathrm{ic}}$, the new prompt is constructed as $\hat{p} = t_{\mathrm{feature}} \oplus p_{\mathrm{task}} \oplus \{(x_{\mathrm{ic}}^k, \hat{y}_{\mathrm{ic}}^k)\}_{k=1}^K$, where $p_{\mathrm{task}}$ is the task instruction, as illustrated in Figure 1. Then $\hat{p}$ would be compared with the current best prompt $p^*$ based on their utility scores as defined in Eq. (1). If $\hat{p}$ scores higher, SICO replaces $p^*$ with it. After $N$ iterations, $p^*$ is returned as the final prompt. More implementation details of SICO are shown in Appendix A.

---

**Algorithm 2** Greedy text optimization (GreedyOPT)

---

**Require:** Text $y$, substitutions $C$ of $y$, proxy detector $\mathcal{P}_{\mathrm{AI}}$
1: $C_{i,*} = \underset{C_{i,j}}{\arg\min}\ \mathcal{P}_{\mathrm{AI}}(y_{(i,j)}), \ \forall y_i \in y$
   where $y_{(i,j)} = \mathrm{SUB}(y_i, C_{i,j})$
2: **for** each $y_i$ in $y$ **do**
3:    $y \leftarrow \mathrm{SUB}(y_i, C_{i,*})$
4: **end for**
5: **return** $y$

---

[2]Content words are the words that carry meanings, consisting of nouns, verbs, adjectives and adverbs.

### 3.3 SICO for Paraphrasing

The approach described above directly generates the task output to evade detectors. We refer to this direct approach as SICO-Gen. Alternatively, SICO can be easily adapted for paraphrasing, which we term as SICO-Para. Instead of direct generation, SICO-Para evades detectors in two steps. Initially, LLM produces an intermediate task output, typically incapable of evading detectors. Then. this output is paraphrased using SICO-Para to successfully evade detectors. Switching from SICO-Gen to SICO-Para requires only two adjustments: (1) the task input $x$ is set to the AI-generated output text in $D$ and $\mathbf{X}_{eval}$; (2) task instruction $p_{\text{task}}$ is modified to paraphrasing instruction.

## 4 Experiments

### 4.1 Experimental Setup

**Tasks & datasets** We consider three real-world tasks that are susceptible to the misuse of LLMs, i.e., academic essay writing (Writing), open-ended question answering (QA), and fake review generation (Review). We use GPT-3.5, one of the most powerful LLMs, to complete the tasks and generate text in our experiments.

For academic writing, we employ Wikipedia paragraphs from SQuAD dataset (Rajpurkar et al., 2016) as human-written text. Following the approach in Mitchell et al. (2023), we use the first 30 words of these paragraphs as task inputs and ask GPT-3.5 to complete the rest. For open-ended question answering, we sample questions from Eli5 (Fan et al., 2019) dataset and ask GPT-3.5 to generate answers, following Krishna et al. (2023). For fake review generation, we first instruct GPT-3.5 to extract the business name and five keywords from human-written reviews from Yelp dataset (Zhang et al., 2015), and then generate fake reviews based on the extracted information with specified sentiment. For each task, we collect 200 examples from GPT-3.5 (called original AI-generated text) and 200 human-written examples from corresponding dataset. More details about dataset can be found in Appendix E.

**Detectors** Six representative detectors belonging to three different types are considered. Details of these detectors can be found in Appendix C.

*Training-based methods.* (i) GPT-3.5 Detector (GPT3-D) (Guo et al., 2023): a RoBERTa model (Liu et al., 2019) fine-tuned on text generated by GPT-3.5. (ii) GPT2 Detector (GPT2-D) (Solaiman et al., 2019): a RoBERTa detector officially released by OpenAI, fine-tuned on GPT2-generated text.

*Statistical methods.* (i) DetectGPT (Mitchell et al., 2023) evaluates the variation in a language model's log probability by introducing minor perturbations to the detected text. (ii) Log-Rank (Mitchell et al., 2023) is a statistical method that employs a language model to compute the mean prediction rank of each token in a text, given its preceding context. We utilize a relatively smaller language model, GPT2-medium (Radford et al., 2019), for both methods. Because Mireshghallah et al. (2023) find that smaller language models have better detection performance than larger ones.

*APIs.*[3] (i) GPTzero (Tian, 2023) is a widely-used commercial detector, cooperated with many academic organizations. (ii) OpenAI Detector (OpenAI-D) (OpenAI, 2023) is officially offered by OpenAI, fine-tuned from a language model.

**Baselines** We consider four paraphrasing baselines that evade detectors by paraphrasing the original AI-generated text. Specifically, two recently proposed methods are considered: (1) *Parrot* (Sadasivan et al., 2023) and (2) *DIPPER* (Krishna et al., 2023). Both methods employ an external neural network specifically trained for paraphrasing. In addition, we include two prompting baselines to instruct GPT-3.5 to paraphrase the original AI-generated text: (3) *GPT-Para* that uses the straightforward instruction *"Paraphrase this"* to assess the capabilities of GPT-3.5 without intricate prompt engineering, and (4) *Human Prompt* that utilizes a human-designed prompt. More details can be found in Appendix A.2.

**Evaluation metrics** We use the area under the ROC curve (AUC) to measure the performance of detectors. The ROC curves are also illustrated to show the detection performance under different classification thresholds.

---

[3]We consider the API versions of May 15, 2023. For OpenAI-D, we follow the implementation of Krishna et al. (2023).

Table 1: AUC scores of detectors on text generated by different methods. "–" refers to the detector's AUC score on the original AI-generated text, without applying any evasion methods. Symbol '*' represents that SICO uses GPT3-D as the proxy detector for prompt construction. For each detector, the lowest AUC score is indicated in **bold**, and the second-lowest is underlined.

| Dataset | Method | GPT3-D* | GPT2-D | GPTzero | OpenAI-D | DetectGPT | Log-Rank |
|---------|--------|---------|--------|---------|----------|-----------|----------|
| Writing | – | 0.908 | 0.848 | 0.779 | 0.789 | 0.834 | 0.914 |
| | Parrot | 0.666 | 0.645 | 0.632 | 0.744 | 0.502 | 0.577 |
| | DIPPER | 0.736 | 0.907 | 0.689 | 0.750 | 0.550 | 0.684 |
| | GPT-Para | 0.879 | 0.623 | 0.631 | 0.690 | 0.569 | 0.713 |
| | Human Prompt | 0.852 | 0.560 | 0.491 | 0.655 | 0.676 | 0.759 |
| | SICO-Para | **0.239** | 0.332 | 0.290 | 0.488 | **0.149** | **0.147** |
| | SICO-Gen | 0.242 | **0.099** | **0.184** | **0.311** | 0.441 | 0.318 |
| QA | – | 0.981 | 0.906 | 0.923 | 0.781 | 0.876 | 0.956 |
| | Parrot | 0.922 | 0.837 | 0.849 | 0.698 | 0.689 | 0.806 |
| | DIPPER | 0.888 | 0.962 | 0.869 | 0.722 | 0.604 | 0.782 |
| | GPT-Para | 0.956 | 0.797 | 0.811 | 0.699 | 0.640 | 0.782 |
| | Human Prompt | 0.912 | 0.625 | 0.791 | 0.656 | 0.662 | 0.757 |
| | SICO-Para | **0.407** | 0.576 | 0.572 | 0.541 | **0.178** | **0.183** |
| | SICO-Gen | 0.668 | **0.489** | **0.494** | **0.524** | 0.497 | 0.535 |
| Review | – | 0.925 | 0.952 | 0.939 | 0.960 | 0.808 | 0.982 |
| | Parrot | 0.871 | 0.934 | 0.913 | 0.882 | 0.654 | 0.893 |
| | DIPPER | 0.875 | 0.984 | 0.888 | 0.824 | 0.515 | 0.814 |
| | GPT-Para | 0.899 | 0.851 | 0.833 | 0.925 | 0.542 | 0.864 |
| | Human Prompt | 0.839 | 0.610 | 0.856 | 0.858 | 0.619 | 0.851 |
| | SICO-Para | 0.465 | **0.264** | 0.599 | **0.540** | **0.270** | **0.300** |
| | SICO-Gen | **0.455** | 0.619 | **0.399** | 0.607 | 0.485 | 0.583 |

For each task, we evaluate AUC score using 200 human-written text and 200 original or paraphrased AI-generated text. For each task input, we run each evasion method *only once*, instead of repeating multiple times until successful evasion, to simulate real-world scenarios where the target detector is inaccessible.

**Experimental settings** We set $|\mathbf{X}_{eval}| = 32$, $K = 8$, $N = 6$, and use GPT-3.5, specifically *gpt-3.5-turbo-0301*, as the LLM, where the inference parameters are kept in default. And we use GPT3-D as the proxy detector. Experiments using other LLMs and proxy detectors are presented in Section 5.2.

## 4.2 Evasion Performance and Analysis

Table 1 presents the performance of SICO and other baselines against six detectors in AUC score. SICO consistently outperforms other baselines by a substantial margin in all cases. Notably, in most cases, SICO reduces the AUC score to less than 0.5, equivalent to the expected performance of a random classifier. Figure 2 shows the ROC curves of evasion methods on academic writing task. One can clearly observe that SICO curves lie below others along different thresholds, often lower than the random classifier curve. More evasion results including ROC cures and detection rates are shown in Appendix G.

One interesting trend is that SICO-Para consistently outperforms SICO-Gen against statistical detectors, i.e., DetectGPT and Log-Rank. We speculate this performance difference comes from the varying influence of the prompt on the generated text between the two methods. In SICO-Para, the distribution of generated text is largely influenced by the original AI-generated text, which is in the prompt. However, in SICO-Gen, the distribution of generated text depends more on the previously generated text. Given that statistical detectors have access to the newly generated text but not the prompt, their estimation of token probability becomes

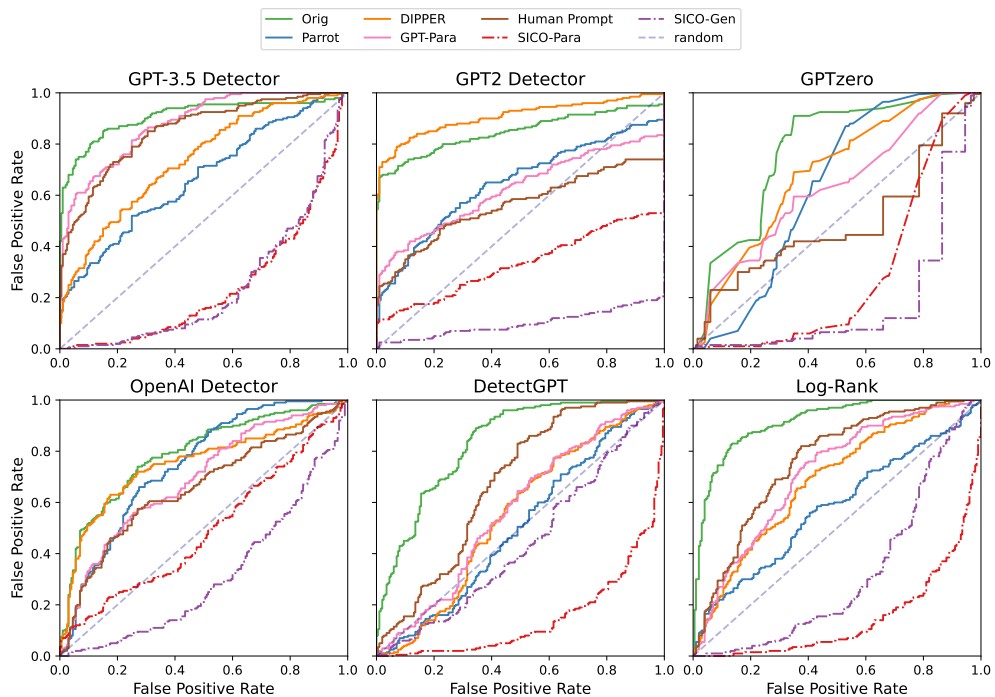

Figure 2: ROC curves for six detectors evaluating text generated by various evasion methods in an academic writing task.

less accurate for SICO-Para text, thus misleading the detection. It might also explain why GPT-Para can reduce the performance of statistical detectors.

### 4.3 Human Evaluation

### 4.3.1 Text Quality

From the users' perspective, using AI-generated text goes beyond evading detection systems; the usability of text is equally critical. For example, for academic writing task, users expect the text to be readable, properly formatted, and relevant to the given topic. Therefore, we evaluate the usability of text based on two criteria: readability and task completion rate. For each task, we randomly sample 200 examples generated by four methods (50 per method), including human-written text. Then we ask three human annotators to rate the readability of text on a scale from 1 to 5, and judge if the text accomplishes the task's goal. More details of human evaluation are shown in Appendix D.

Table 2: Human evaluation results. "Avg.D." represents the average difference between the results achieved by the evasion method and the results achieved by human-written text on the three tasks. The best value of each task is set **bold**.

| Method | Readability ↑ | | | | Task Completion Rate % ↑ | | | |
|---|---|---|---|---|---|---|---|---|
| | Writing | QA | Review | Avg.D. ↑ | Writing | QA | Review | Avg.D. ↑ |
| DIPPER | 3.52 | 4.12 | 3.42 | -0.27 | 70.6 | 100 | 61.6 | -13.3 |
| SICO-Para | 3.68 | **4.36** | 3.58 | -0.09 | 82.0 | 100 | 72.4 | -5.9 |
| SICO-Gen | 3.84 | 4.28 | **3.70** | **-0.02** | 93.6 | 100 | 69.6 | **-2.9** |
| Human-Written | **3.92** | **4.36** | 3.60 | - | **98.2** | 100 | **73.8** | - |

As shown in Table 2, both SICO-Gen and SICO-Para demonstrate superior performance over DIPPER in terms of task completion and readability over three tasks. Furthermore, SICO-generated text performs competitively compared with human-written text in both metrics, with a negligible difference less than 0.1. In contrast, DIPPER exhibits inferior performance relative to human-written text, particularly with a notable 0.27 decline in readability.

### 4.4 Imperceptibility

Another key factor for the usability of AI-generated text is its imperceptibility. If text is easily identified as AI-generated by humans, its usability might be significantly influenced. Thus, we conducted an experiment to assess the imperceptibility of text. We sampled 200 examples (50 each from AI, DIPPER, SICO-Gen, and human) across three tasks. Three human annotators were asked to identify whether the text was AI-generated or human-written. All annotators in the experiment used ChatGPT before. The values in Table 3 show the percentage of texts recognized as AI-generated by the annotators, where a lower percentage indicates higher imperceptibility. The results demonstrate that SICO remarkably reduces the probability of being recognized by annotators for QA and review generation tasks.

Table 3: Imperceptibility of text from different sources.

| Source | Writing | QA | Review |
|--------|---------|-----|--------|
| AI | 24% | 52% | 54% |
| DIPPER | 31% | 46% | 66% |
| SICO | 22% | 14% | 37% |
| Human | 18% | 13% | 22% |

For the academic writing task, the similar low detection rates for AI and human-generated text (24% VS 18%) annotators' detection capabilities are less accurate for this task, explaining why SICO is less effective here. More details can be found in Appendix D.

### 4.5 Cost Efficiency

In terms of the data prerequisite, SICO only needs $K + |\mathbf{X}_{eval}|$ human-written input-output examples to build prompt, which is $8 + 32 = 40$ in the experiments. The other AI-generated text can be produced by LLM leveraging these human samples. Furthermore, SICO offers the advantage of low cost for prompt construction. Based on three repeated runs, the actual USD costs of SICO-Para are $1.04 \pm 0.04$, $1.08 \pm 0.05$, and $0.75 \pm 0.04$ for Writing, QA, Review tasks, respectively.

## 5 Further Experiments

### 5.1 Ablation Study

We conducted an ablation study over academic writing task to to evaluate the contribution of individual components within the SICO framework. "Human-ICE" denotes the approach where human-written text is directly utilized as the in-context example for constructing the prompt. "w/o feature" and "w/o ICE" refer to the prompts without feature text and the optimized in-context examples, respectively. "w/o OPT" represents the initial prompt before optimization (see Line 3 in Algorithm 1). In our experiment, we explore SICO-Para on three types of detectors: GPT3-D, OpenAI-D and DetectGPT.

Results in Table 4 shows that directly using human-written text is ineffective, even making the detection more accurate. We speculate that the human-written examples are too heterogeneous and characterized in multiple ways, so LLM cannot effectively learn their attributes. Besides, the importance of feature text is comparatively less than that of optimized in-context examples. Furthermore, the result reveals the significant role of the optimization step in SICO. Using in-context examples that are not optimized is essentially equivalent to not using any in-context examples.

### 5.2 SICO with Different Proxy Detectors and LLMs

As described in Section 3, SICO requires a proxy detector and a LLM to construct a prompt. In this experiment, we explore the performance of SICO-Para on writing task, using three types of proxy detectors: (1) training-based model GPT-3.5 detector, (2) API detector GPTzero, and (3) statistical method DetectGPT.

Table 4: The AUC scores of detectors on text generated by different methods. "–" indicates the case where no evasion method is used. 'AVG' represents the average AUC scores across detectors.

| Method | GPT3-D | OpenAI-D | DetectGPT | AVG |
|---|---|---|---|---|
| – | 0.908 | 0.789 | 0.834 | 0.844 |
| Human-ICE | 0.918 | 0.825 | 0.847 | 0.863 |
| SICO-Para | 0.239 | 0.488 | 0.149 | 0.292 |
| w/o feature | +0.106 | +0.072 | +0.051 | +0.076 |
| w/o ICE | +0.359 | +0.133 | +0.411 | +0.301 |
| w/o OPT | +0.361 | +0.154 | +0.364 | +0.293 |

For different LLMs, we adopt Vicuna-13B (Chiang et al., 2023), an open-source chatbot fine-tuned from LLaMa (Touvron et al., 2023). Results in Table 5 show that SICO maintains a high degree of detection evasion performance, regardless of proxy detectors or LLMs. In most cases, SICO manages to reduce the AUC of detectors by approximately 0.4. More results of using different LLMs can be found in Appendix B.4.

Table 5: The AUC scores of SICO using different proxy detectors and LLMs on writing task. The first line indicates the performance without applying any evasion method.

| LLM | Proxy Detector | GPT3-D | GPT2-D | GPTzero | OpenAI-D | DetectGPT | Log-Rank |
|---|---|---|---|---|---|---|---|
| - | - | 0.908 | 0.848 | 0.779 | 0.789 | 0.834 | 0.914 |
| GPT-3.5 | GPT3-D | 0.239 | 0.332 | 0.290 | 0.488 | 0.149 | 0.147 |
| GPT-3.5 | GPTzero | 0.549 | 0.143 | 0.299 | 0.458 | 0.429 | 0.493 |
| GPT-3.5 | DetectGPT | 0.291 | 0.361 | 0.244 | 0.483 | 0.174 | 0.173 |
| Vicuna-13B | GPT3-D | 0.393 | 0.777 | 0.380 | 0.690 | 0.337 | 0.375 |

### 5.3 Examples of SICO text

Table 6: Fake reviews generated by SICO. The first line shows the task input of review generation.

| **Object**: Sushi Station **Keywords**: revolving sushi bar, affordable, krab mix, creamy sauce, fresh menu. **Sentiment**: Positive | $\mathcal{P}_{AI}$ |
|---|---|
| **AI**: If you're a sushi lover, you have to check out Sushi Station! This revolving sushi bar is not only affordable, but it offers a fresh and delicious menu that will leave you craving more. My personal favorite is the krab mix sushi with its creamy sauce - it's simply heavenly. The staff is friendly and attentive, and the atmosphere is perfect for a casual night out with friends or family. Don't hesitate to give Sushi Station a try - you won't be disappointed! | 0.99 |
| **SICO-Para**: Sushi enthusiasts, please take note: Sushi Station provides an unbeatable revolving sushi bar experience at wallet-friendly prices. Their menu is of the freshest quality and tastes nothing short of delicious. My top pick has to be the krab mix sushi; its creamy sauce is pure bliss. Additionally, their friendly staff provided excellent service, while the casual ambiance proved ideal for a relaxed night out with friends or family. Don't hesitate to give Sushi Station a try, you won't regret it! | 0.00 |
| **SICO-Gen**: I recently tried out Sushi Station's revolving sushi bar and I must say, it was an amazing experience. The affordable pricing along with the wonderful krab mix and creamy sauce they offer is what really made it stand out from the rest. The menu was constantly updated with fresh and delicious options making it hard to choose just one. If you're looking for a great sushi experience, Sushi Station is definitely worth a visit. | 0.04 |

Table 6 presents the fake reviews generated by SICO-Gen and SICO-Para. The generated text shows high readability and fulfill the task's goal, successfully mentioning all keywords and generating positive reviews of the specified object. The AI probability, denoted as $\mathcal{P}_{AI}$ in the table, is determined by the GPT3-D. More examples are shown in Appendix H.

# 6 Defensive Methods for SICO

## 6.1 Training Detectors with SICO Data

To defend against SICO, a straightforward approach is to train a detector on SICO text. We proposed an ensemble method, which utilizes two detectors trained separately to identify original AI-generated text and SICO text. The final AI probability for an input is calculated using the highest one from the two detectors, based on the assumption that both detectors should assign a low probability to human-written text. We avoided training a single detector with SICO text augmentation because our experiments indicated that this approach reduces the model's ability to identify the original AI-generated text.

To test the effectiveness of this method, we obtain the SICO detector by fine-tuning a RoBERTa model with 5k SICO examples and 5k human-written examples. The human-written examples are sampled from WebText training set (Radford et al., 2019). The SICO examples are generated by using three different SICO-Para prompts to rewrite AI-generated text from the GPT2 output dataset (Solaiman et al., 2019). The SICO prompts used for detector training is entirely different from the one used for attack. We adopted the "GPT2-D" as the original AI-generated text detector. The results in Table 7 show that the ensemble approach significantly improves detection capabilities against SICO text, while maintaining the ability to identify the original AI-generated text. We believe this approach can be easily adapted to defend against other types of attacks, such as DIPPER, by training and incorporating detectors with adversarial attack.

Table 7: AUC scores of ensemble detector against normal AI-generated text and SICO text. "AI" refers to the detector's AUC score on the original AI-generated text.

| Dataset | Detector | AI | SICO-Para | SICO-Gen |
|---------|----------|-----|-----------|----------|
| Writing | GPT2-D | 0.848 | 0.332 | 0.099 |
|         | + SICO-D | $0.842_{(-0.006)}$ | $0.762_{(+0.430)}$ | $0.691_{(+0.592)}$ |
| QA      | GPT2-D | 0.906 | 0.576 | 0.489 |
|         | + SICO-D | $0.892_{(-0.014)}$ | $0.815_{(+0.239)}$ | $0.784_{(+0.295)}$ |
| Review  | GPT2-D | 0.952 | 0.264 | 0.619 |
|         | + SICO-D | $0.949_{(-0.003)}$ | $0.782_{(+0.518)}$ | $0.834_{(+0.215)}$ |

## 6.2 Discussion of AI Detection Arms Race

Considering the evolution of AI detection techniques, the attackers and defenders are actually engaged in an arms race, where both sides continuously develop their own optimal strategies of variability and sophistication to overcome the opponents. We expect the newly proposed attack method will remain effective until the defender collects enough adversarial examples and trains a new detector on them. Then the attacker might develop a new technique to evade the most recent detector. This arms race presupposes that both sides are willing to share information like attack technique or detector API. If either side stops sharing, the race will come to a halt. Practically speaking, the defenders hold a superior position in this race, as they can restrict the access to detector, thereby preventing the attacker from improving their methods. Moreover, the defender, typically a large company, possesses more resources, including financial and computing power. The arms race represents the ultimate question in this field, which is complex and significant. Hence, we only provided a surface-level discussion. A more thorough investigation and experimentation on it are left for future research.

## 7 Conclusion

In conclusion, we have proposed a novel in-context learning approach, SICO, designed to guide LLMs in generating text that can effectively evade detectors. Our extensive experiments on evasion demonstrate the superior performance of SICO, which significantly reduces the detection capabilities of existing AI text detectors across three tasks. A comprehensive human evaluation shows SICO text can achieve human-level readability and task completion rates.

Looking ahead, SICO could act as a data generator and be integrated during the training phase of AI detectors, which may enhance their robustness. Furthermore, the core concept of SICO, namely, substitution-based in-context learning, could be applied to a variety of text generation tasks. We believe that this opens up new avenues for future research in the fields of text generation and in-context learning.

## 8 Ethics Statement

The intention of this paper is not to offer a potential method for evading AI-generated text detection systems. Instead, our aim is to raise awareness within the broader community about the vulnerabilities of existing AI-generated text detection systems to such technology. As many LLMs are public available and free to use, many people can adjust their prompt and generate text that evades these detectors. Given the ease of evasion illustrated in this study, these detectors are not robust yet. We hope the research community can stress test their detectors against text generated by carefully crafted prompt, and create more robust detectors in the future.

Besides presenting a potent attack technique, we also offer defensive methods against it. We believe future research will develop more sophisticated methods to enhance the robustness of AI detectors. To support the research in this field, we make our codes and data publicly available.

### Acknowledgments

This work was supported by the National Key Research and Development Program of China under Grant 2022YFA1004102, and the Guangdong Major Project of Basic and Applied Basic Research (Grant No. 2023B0303000010).

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

# A  Implementation Details

## A.1  SICO

### A.1.1  Feature extraction

In feature extraction step, we instruct LLM to extract 5 features and calculate the utility score $\mathcal{U}$ of prompts encompassing each of these features. Then we select the feature with the highest utility for further steps. The goal of this step is to find a good feature to accelerate process, and make the whole process stable. Because sometimes LLM cannot extract useful features to evade detectors. The pseudo-code illustrating this selection process is outlined in Algorithm 3. Table 8 presents the prompt template used for feature extraction. Here, $K$ text pairs generated by AI and Human are positioned within their respective positions. Table 21 shows the examples for feature extracted by LLM.

---

**Algorithm 3** Feature selections

---

**Require:** list of features $T_{\text{feature}}$, prompt utility function $\mathcal{U}(\cdot)$
 1: Initialize $t^*_{\text{feature}} \leftarrow \emptyset$
 2: Initialize $\mathcal{U}_{max} \leftarrow -\infty$
 3: **for** each feature $t_{\text{feature},i}$ in $T_{\text{feature}}$ **do**
 4:     Construct prompt $p_i \leftarrow t_{\text{feature},i} \oplus p_{\text{task}}$
 5:     **if** $\mathcal{U}(p_i) > \mathcal{U}_{max}$ **then**
 6:         $t^*_{\text{feature}} \leftarrow t_{\text{feature},i}$
 7:         $\mathcal{U}_{max} \leftarrow \mathcal{U}(p_i)$
 8:     **end if**
 9: **end for**
10: **return** $t^*_{\text{feature}}$

---

Table 8: Prompt for feature extraction.

---

Here are the writings from AI and human:
AI writing: <AI-generated text>
Human writing: <human-written text>
...
What is the key distinct feature of human's writings?
[LLM complete]

---

**LLM consistently extract useful features.**    To test if LLM can reliably extract useful features, we conducted three separated experiments by running three feature extractions on different sets of human-written text and AI-generated text. We use three different extracted features to guide LLM generation and test the AUC drop after adopting 3 different features compared with the originally generated text on the Writing task. Table 9 shows the results, indicating that LLM consistently extracts useful features for detector evasion from different examples.

Table 9: AUC drop of different features extracted based on different human-written and AI-generated text.

| Orig. | Feature 1 | Feature 2 | Feature 3 |
|-------|-----------|-----------|-----------|
| 0.908 | -0.288 | -0.261 | -0.142 |

### A.1.2  Task instructions

Table 10 shows the actual task instruction $p_{\text{task}}$ we used in SICO. As mentioned in Section 3, feature text $t_{\text{feature}}$ at first step will be inserted before these task instructions. The "Paraphrase" instruction is $p_{\text{para}}$ used in paraphrase generation for substitution (Line 6 of Algorithm 1).

### A.1.3  Word Substitution

We employ WordNet synsets to derive synonyms for given words. During optimization of in-context examples, we only substitute content words, namely nouns, verbs, adjectives, and adverbs. Furthermore, we part-of-speech (POS) tag of the synonym to ensure it aligns with the original word. For POS tagging, we utilize the Stanford POS Tagger (Toutanova et al., 2003). Additionally, to maintain fluency in the modified text after substitution, we employed a pretrained mask language model to exclude synonyms with low likelihood. In experiment we use RoBERTa-base model (Liu et al., 2019).

Table 10: Task instructions of each task.

| Task | Task instruction $p_{\text{task}}$ |
|---|---|
| Writing | Based on the description, complete an academic paragraph in human style writings:
Prompt: *< task input >*
Human: [LLM complete] |
| QA | Based on the description, answer questions in human style writings:
Q: *< task input >*
Human: [LLM complete] |
| Review | Based on the description, write a human review about the given object and keywords, with a specified sentiment:
Object, Keywords, Sentiment: *< task input >*
Human: [LLM complete] |
| Paraphrase | Based on the description, rewrite this in human-style writings:
Origin: *< original AI-generated text >*
Human: [LLM complete] |

### A.2  Baselines

### A.2.1  DIPPER

We choose the best evasion performance parameter setting from the original paper (Krishna et al., 2023), which is 60 for lexical diversity and 60 for re-ordering. And we set sampling temperature to 0.75, following the original implementation.

### A.2.2  Human prompt

We carefully design a paraphrase prompt based on the detection idea of GPTzero (Tian, 2023) and prompt shared online (Uploader, 2023), which distinguish the AI-generated content from Human-written by *perplexity* and *burstiness*, stated by its creator[4]. *Perplexity* is the concept raised in NLP field, which measures how well a language model predicts a text sample. A text with a lower perplexity score indicates that the language model is better at calculating the next word that is likely to occur in a given sequence. On the other hand, *burtiness* basically measures the variation between sentences, including sentence length and structures. The lower the values for these two factors, the more likely it is that a text was produced by an AI. Table 11 shows the prompt we designed.

---

[4]https://theconversation.com/we-pitted-chatgpt-against-tools-for-detecting-ai-written-text-and-the-results-are-troubling-199774

Table 11: Human-designed prompt to evade AI-generated text detection.

When it comes to writing content, two factors are crucial, "perplexity" and "burstiness". Perplexity measures the complexity of text. Separately, burstiness compares the variations of sentences. Humans tend to write with greater burstiness, for example, with some longer or complex sentences alongside shorter ones. AI sentences tend to be more uniform.

Paraphrase the following AI sentence to be human-like, with a good amount of perplexity and burstiness:
Orig: < *orignal AI-generated text* >
New: [LLM complete]

## B   Extra Experiments

### B.1   Semantic preserving

We measure semantic similarity using t5-based sentence encoder (Ni et al., 2022), which leads in semantic text similarity task of MTEB benchmark (Muennighoff et al., 2023). Table 12 reports a comparison of the cosine similarity of text before and after paraphrasing by different methods. Our methods successfully preserves the

Table 12: Cosine similarity between original AI-generated text and their respective paraphrased versions using different methods. The best scores in each task are highlighted in **bold**.

| Method | Writing | QA | Review |
|---|---|---|---|
| Parrot | 0.964 | 0.961 | 0.966 |
| DIPPER | 0.956 | 0.941 | 0.940 |
| GPT-Para | **0.986** | **0.982** | **0.987** |
| Human Prompt | 0.979 | 0.968 | 0.978 |
| SICO-Para | 0.976 | 0.964 | 0.971 |

semantic meaning during paraphrasing, and beats the other specifically trained paraphraser. Paraphrasing directly using GPT-3.5 yields the most promising results.

### B.2   Stability of SICO

SICO is able to consistently construct effective detection evasion prompts, irrespective of the diversity in the initial AI-human text pairs and randomized samples drawn from the LLMs. This demonstrates the effectiveness of SICO in various initial conditions and settings, highlighting its applicability to diverse scenarios. Figure 3 presents the detection evasion performance of the best prompt at each step, denoted as $\mathcal{U}(p^*)$ in Equation 1, where a higher value means higher evasion performance. The prompt is evaluated by 50 validation examples. We run three distinct experiment for each task to draw the plot. SICO successfully optimizes the initial prompt (at step 0) and achieves a high level of evasion performance across all three tasks, with different in-context examples.

### B.3   SICO performs better against more capable detectors

We use the detectors' performance on the original AI-generated text to represent their base capability. The SICO advancing performance is measured by the AUC difference between best of SICO-Para or SICO-Gen and the best-performing paraphraser baselines. The statistical Pearson correlation is 0.47 with a p-value of 0.048, indicating a moderate positive correlation.

### B.4   SICO with different LLMs

To examine the effectiveness of SICO employing different LLMs, we adopted an additional experiment which employs SICO-Para on three different LLMs: WizardLM-13B Xu et al. (2023), GPT-3.5-turbo-0301, and

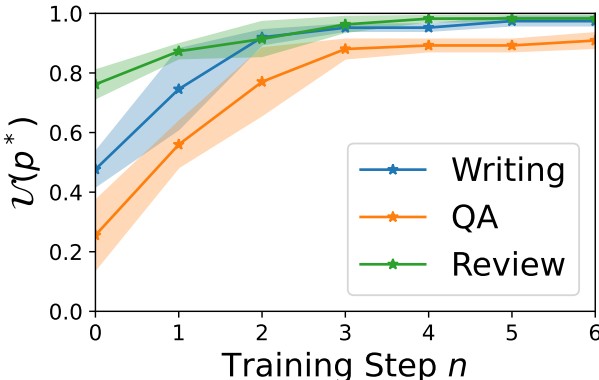

Figure 3: The trajectory of the $\mathcal{U}(p^*)$ during prompt optimization. This plot is derived from three distinct training runs on three tasks.

GPT-4-0613 OpenAI (2022). We employ Chat-D as proxy detector. The results in Table 13 indicate that SICO with different LLMs maintains a high degree of detection evasion performance.

Table 13: The AUC scores of SICO using different LLMs. "-" indicates the performance without applying any evasion method. Symbol "*" represents that SICO uses GPT3-D as the proxy detector for prompt construction.

| Dataset | LLM | GPT3-D* | GPT2-D | DetectGPT | Log-Rank |
|---------|-----|---------|--------|-----------|----------|
| Writing | – | 0.908 | 0.848 | 0.834 | 0.914 |
| | WizardLM | 0.571 | 0.582 | 0.510 | 0.481 |
| | GPT-3.5 | 0.239 | 0.332 | 0.149 | 0.147 |
| | GPT-4 | 0.157 | 0.260 | 0.263 | 0.277 |
| QA | – | 0.981 | 0.906 | 0.876 | 0.956 |
| | WizardLM | 0.721 | 0.467 | 0.561 | 0.404 |
| | GPT-3.5 | 0.402 | 0.566 | 0.178 | 0.183 |
| | GPT-4 | 0.414 | 0.369 | 0.344 | 0.287 |
| Review | – | 0.925 | 0.952 | 0.808 | 0.982 |
| | WizardLM | 0.707 | 0.402 | 0.570 | 0.542 |
| | GPT-3.5 | 0.465 | 0.264 | 0.270 | 0.300 |
| | GPT-4 | 0.408 | 0.307 | 0.427 | 0.382 |

## C    Detectors

In this section, we introduce the mechanism and settings of the detectors in our experiments.

### C.1    GPT-3.5 Detector

GPT-3.5 detector is trained on Human ChatGPT Comparison Corpus (HC3) dataset (Guo et al., 2023), which including answers generated by ChatGPT and human. English-version of HC3 dataset contains five splits: reddit-eli5, OpenQA, Wikipedia, medicine and finance. The base model is RoBERTa-base and we use the model that only take answers as input.

### C.2 GPT2 Detector

GPT-2 detector is obtained by fine-tuning a RoBERTa model with the outputs of the 1.5B-parameter GPT-2 model. The detector and the GPT-2 output dataset are both provided by OpenAI. Although it is trained on GPT-2 outputs, our experiments shows that it can effectively detect text from GPT-3.5.

### C.3 DetectGPT

DetectGPT identifies if a text is generated by a model by observing a unique characteristic: AI-generated text tends to be in areas where the language model's log probability function has a negative curve. Here's how it works: It first perturbs the input text and constructs multiple perturbations of input text. The perturb step is completed by a mask language model. Then it checks the log probability of these variations against the original text by a inner language model. Finally, the text is considered AI-generated if the log probability of the original input text is significantly higher than the log probability of perturbations.

We use z-score implementation of DetectGPT and set sample number to 100 and replacement ratio to 0.3. The inner language model is GPT2-medium and the mask language model is t5-large (Raffel et al., 2020).

### C.4 Log-Rank

Log-Rank method employs the mean prediction rank of each token in a text. Specifically, for each word in a text, given its previous context, it can calculate the absolute rank of this word by an inner language model. Then, for a given text, we compute the score of the text by averaging the rank value of each word. Note that a smaller score denotes the text is more likely to be machine-generated. In experiment, we use GPT2-medium to calculate the rank of tokens to align with the implementation of DetectGPT.

### C.5 GPTzero

GPTzero is a recently proposed commercial detector, employed by many users and oragnizations. As claimed in its websites[5], GPTzero can be used to detect the outputs from detect ChatGPT, GPT4, Bard, LLaMa, and other AI models. GPTZero is the leading AI detector for checking whether a document was written by a large language model such as ChatGPT. GPTZero detects AI on sentence, paragraph, and document level. GPTzero was trained on a large, diverse corpus of human-written and AI-generated text, with a focus on English prose. GPTZero has served over 2.5 million users around the world, and works with over 100 organizations in education, hiring, publishing, legal, and more.

### C.6 OpenAI Detector

OpenAI detector is officially provided by OpenAI after the release of ChatGPT. Although it only offers a web interface, we adopted the API implementation from (Krishna et al., 2023), which uses "model-detect-v2" in the OpenAI API. Through reverse engineering of the website, we determined that the web interfaces indeed call this model.

On July 20, 2023, OpenAI discontinued this detector, "due to its low rate of accuracy."[6]. Considering the discontinuation of the OpenAI detector aligns with our findings, we choose to present the results of it in our paper, though it is out of date.

## D Human Evaluation Details

For each text, we show two questions for the human annotator. In terms of readability, we present the human annotator five options, with the scale of 1 to 5. The higher the value, the more readable of the presented texts. The question is identical for three tasks. The actual question and options are presented in Table 14. For task completion rate, we design three task-specific questions, as show in Table 15. Figure 5 shows the

---

[5]https://gptzero.me/
[6]https://openai.com/blog/new-ai-classifier-for-indicating-ai-written-text

interface of our annotation platform. We estimated that the evaluation time of each text ranges from 60 to 90 seconds. For the imperceptibility experiment, we present the annotators with text and a question to determine if ChatGPT generates the text. All annotators in the experiment used ChatGPT before.

Table 14: Question and options designed for readability.

| Question | On a scale of 1-5, how would you rate the readability of the given text? |
|---|---|
| Options | 1: Very difficult to read and understand.
2: Difficult to read, need extra time to understand.
3: Neutral.
4: Easy to read.
5: Very clear and easy to read. |

Table 15: Question and options designed for task completion.

| Writing | |
|---|---|
| Question | Is this academic essay correctly written (No errors and in academic style)? |
| Options | No, it has some obvious errors or not in academic format. |
| | Yes, it is correctly written. |
| **QA** | |
| Question | Does the answer relate to the question? (regardless of correctness) |
| Options | No, not relative. |
| | Yes, it is relative. |
| **Review** | |
| Question | Does this review impact your decision to choose this service? |
| Options | No influence. |
| | Yes, it provides useful information. |

**Human evaluation on Parrot.** As Parrot method performs better than DIPPER in the Writing task, we conducted a small experiment to evaluate the usability of text generated text, similar to Table 2. We randomly sampled 120 examples (40 for Parrot, SICO-Para and SICO-Gen) from the writing task and asked two human annotators to evaluate them. The experiment result in Table 16 shows that SICO still outperforms Parrot by a large margin.

Table 16: Human evaluation results of Parrot and SICO.

| Method | Readablity | Task Compeletion Rate |
|---|---|---|
| Parrot | 3.35 | 70.0 |
| SICO-Para | 3.80 | 82.5 |
| SICO-Gen | **3.90** | **85.0** |

# E   Datasets

Table 17 presents the prompts we employed to create the initial AI-generated text $y_{\text{AI}}$. For academic writing, we sample Wikipedia paragraphs from SQuAD dataset. Then we give GPT-3.5 the first 30 words of these paragraphs and ask GPT-3.5 to complete the rest.

For open-ended question answering, we sample questions from Eli5 dataset and ask GPT-3.5 to generate answers.

For fake review generation, we first instruct GPT-3.5 to extract the business name and five keywords from human-written reviews from Yelp dataset, and then generate fake reviews based on the extracted information with specified sentiment.

Table 17: Prompt for dataset creation.

| Task | Prompt |
|---|---|
| Writing | Complete academic paragraph with prompt: < *first 30 words of paragraph* > |
| QA | Answer questions: < *question* > |
| Rev-Ext | Review: < *human-writtent review* >
Show the review's object and 5 key words: |
| Rev-Gen | Write a review about given object and key words, with specified sentiment:
< *output of Rev-Ext* >
Review: |

The statistics of three datasets are shown in Table 18

Table 18: Average character length of human-writteng text and the original AI-generated text.

|  | SQuAD | Eli5 | Yelp |
|---|---|---|---|
| Human | 770.3 | 794.7 | 834.3 |
| AI | 796.5 | 580.9 | 505.1 |

# F   Evade Watermarking Detection

SICO-Para can also be utilized to evade watermark detection, similarly to paraphrase approach. The watermarking algorithm we applied was introduced by Kirchenbauer et al. (2023), which only requires access to the LLM's logits at each time step to add watermarks. This algorithm operates in three steps:

1. Mark a random subset of the vocabulary as "green tokens"using the hash of the previously generated token as a random seed.

2. Increase the logit value for every green token by a constant, which denotes the watermark strength.

3. Sample sequences using decoding algorithms.

Verification of this watermark is achievable with blackbox access to the LM and knowledge of the hash function, achieved by tokenizing the text, calculating the standard normal score (z-score) through a hypothesis test, and comparing the observed proportion of green tokens to the expected proportion.

In our experiments, we used the text generated by a watermarked GPT-2, provided by Krishna et al. (2023). We employed the GPT-3.5 detector as proxy detector for training. The AUC and and the detection accuracy associated with various paraphrasing methods are presented in Table 19, where the threshold is set to 2.2 for accuracy measurement.

The results reveal that SICO-Para significantly outperforms other paraphrase techniques in evading watermark detection. Notably, both the AUC score and detection accuracy of SICO-Para are lower than that of other methods.

As we mentioned in Section 2, our work is focused on the vulnerability of training-based and statistical detectors. Therefore, we designed SICO to generate human-like text aimed at fooling these two types

Table 19: Performance of paraphrase methods on watermarking detection.

| Method | AUC | Accuracy |
|--------|-----|----------|
| AI | 0.998 | 99.0% |
| Parrot | 0.980 | 88.5% |
| DIPPER | 0.844 | 33.0% |
| GPT-Para | 0.753 | 18.0% |
| SICO-Para | **0.669** | **14.0%** |
| Human | - | 1.0% |

of detectors, which differentiate between human and AI-generated text based on language characteristics. However, as discussed in the previous section, watermarking methods distinguish human and AI-generated text by adding unnoticeable mathematical character in AI-generated text. Theoretically, any paraphrasing method that can significantly rewrite the original text will remove the watermark, thus evading the detector. For example, even the simplest baseline model, "GPT-Para", proved to be effective at evading the watermark detection, reducing accuracy from 99% to 18%. Based on the aforementioned reasons, we put the watermark results in the appendix, though SICO-para can also help evade watermark detectors.

## G  Evasion Performance

### G.1  ROC Curves

Figure 2 shows ROC curves of different detectors presented with text generated by different methods, on open-ended question answering and fake review generation task. SICO curves lie below other baselines.

### G.2  Detection Accuracy

Given that detection rates highly depend on the selected detection threshold, we establish two thresholds for each detector. The high threshold fixes the *false positive rate* (FPR) at a low level of 0.05, which means only 5% of human-written text will be classified as AI-generated. The low threshold fixes the *true positive rate* (TPR) at a high level of 0.9, based on the original AI-generated text. In this case, 90% of original AI-generated text will be correctly classified. Table 20 shows the detection accuracy on three task. In comparison with other paraphrasing methods, SICO yields the lowest detection rates in most cases.

## H  Examples of Text Generated by SICO

The examples of text generated by SICO across three tasks are presented in Tabel 22-24.

Table 20: Detection accuracy on three tasks. "AI" refers to the detection rate on the original AI-generated text. The lowest score of each detector is indicated in **bold**, and second-lowest is underlined.

**Writing**

| Method | High Threshold | | | | | | Low Threshold | | | | | |
|---|---|---|---|---|---|---|---|---|---|---|---|---|
| | GPT3-D | GPT2-D | GPTzero | OpenAI-D | DetectGPT | Log-Rank | GPT3-D | GPT2-D | GPTzero | OpenAI-D | DetectGPT | Log-Rank |
| AI | 71.0% | 68.5% | 6.5% | 33.5% | 32.0% | 62.0% | 90.0% | 90.0% | 86.5% | 90.0% | 90.0% | 90.0% |
| Parrot | 25.0% | 25.5% | 0.5% | 12.5% | 2.5% | 16.5% | 53.5% | 77.0% | 46.5% | 93.0% | 33.5% | 36.5% |
| DIPPER | 28.5% | 74.5% | 2.5% | 33.0% | 1.0% | 14.0% | 62.5% | 95.0% | 64.5% | 82.0% | 44.0% | 51.0% |
| GPT-Para | 58.5% | 35.0% | 5.0% | 14.5% | 5.0% | 14.0% | 85.5% | 73.0% | 54.0% | 84.0% | 48.0% | 56.5% |
| Human Prompt | 49.0% | 26.5% | 10.5% | 15.5% | 10.5% | 18.0% | 84.0% | 63.5% | 40.0% | 76.0% | 64.0% | 67.0% |
| SICO-Para | 1.0% | 12.5% | **0.0%** | 11.5% | **0.0%** | **0.0%** | **5.5%** | 40.5% | 4.0% | 56.5% | **3.5%** | **2.0%** |
| SICO-Gen | **0.5%** | **2.5%** | 1.5% | **2.0%** | 3.0% | 1.0% | **6.0%** | **12.0%** | **3.5%** | **32.0%** | 27.5% | 5.5% |
| Human | 5.0% | 5.0% | 4.0% | 5.0% | 5.0% | 5.0% | 30.5% | 64.5% | 34.5% | 61.0% | 38.5% | 28.5% |

**QA**

| Method | High Threshold | | | | | | Low Threshold | | | | | |
|---|---|---|---|---|---|---|---|---|---|---|---|---|
| | GPT3-D | GPT2-D | GPTzero | OpenAI-D | DetectGPT | Log-Rank | GPT3-D | GPT2-D | GPTzero | OpenAI-D | DetectGPT | Log-Rank |
| AI | 91.0% | 70.0% | 70.0% | 30.5% | 40.5% | 78.5% | 90.0% | 90.0% | 89.5% | 90.0% | 90.0% | 90.0% |
| Parrot | 62.0% | 49.0% | 42.5% | 13.5% | 7.0% | 34.5% | 58.0% | 78.0% | 75.0% | 85.5% | 62.5% | 52.5% |
| DIPPER | 43.5% | 86.5% | 53.0% | 22.0% | 7.5% | 27.5% | 38.5% | 96.5% | 82.0% | 84.0% | 46.5% | 38.5% |
| GPT-Para | 75.5% | 46.5% | 42.0% | 14.0% | 7.0% | 32.5% | 73.0% | 77.5% | 61.0% | 86.5% | 55.0% | 45.0% |
| Human Prompt | 56.0% | 32.0% | 42.5% | **12.5%** | 10.5% | 31.0% | 51.0% | 56.5% | 61.0% | 83.0% | 57.0% | 47.5% |
| SICO-Para | **0.0%** | 27.0% | **0.5%** | **12.5%** | **0.0%** | **0.0%** | **0.0%** | 51.0% | **6.5%** | 63.5% | **1.0%** | **0.0%** |
| SICO-Gen | 19.5% | **24.5%** | 21.0% | **12.5%** | 20.0% | 26.0% | 17.0% | **47.5%** | 36.0% | **61.0%** | 45.5% | 37.0% |
| Human | 5.0% | 5.0% | 5.0% | 5.0% | 5.0% | 5.0% | 4.5% | 29.0% | 24.0% | 61.5% | 33.5% | 11.5% |

**Review**

| Method | High Threshold | | | | | | Low Threshold | | | | | |
|---|---|---|---|---|---|---|---|---|---|---|---|---|
| | GPT3-D | GPT2-D | GPTzero | OpenAI-D | DetectGPT | Log-Rank | GPT3-D | GPT2-D | GPTzero | OpenAI-D | DetectGPT | Log-Rank |
| AI | 55.0% | 86.5% | 76.5% | 84.5% | 25.5% | 90.5% | 90.0% | 90.0% | 87.5% | 90.0% | 90.0% | 90.0% |
| Parrot | 34.0% | 78.0% | 59.5% | 47.0% | 7.0% | 64.0% | 79.0% | 85.5% | 79.0% | 60.0% | 70.5% | 63.5% |
| DIPPER | 37.0% | 96.5% | 56.5% | 56.0% | 3.5% | 45.5% | 82.5% | 98.0% | 73.5% | 62.0% | 45.0% | 44.0% |
| GPT-Para | 46.0% | 71.5% | 36.5% | 62.0% | 3.5% | 53.0% | 87.0% | 74.5% | 52.0% | 79.5% | 54.5% | 50.0% |
| Human Prompt | 32.0% | 40.0% | 51.0% | 47.5% | 10.0% | 47.0% | 69.0% | 43.0% | 63.5% | 62.0% | 63.0% | 45.5% |
| SICO-Para | **1.5%** | **11.5%** | 3.5% | **9.0%** | **2.0%** | **2.5%** | **12.5%** | **15.0%** | 14.0% | **12.0%** | **16.0%** | **2.5%** |
| SICO-Gen | 11.0% | 39.5% | **0.5%** | 14.0% | 11.5% | 19.0% | 28.0% | 42.5% | **0.5%** | 20.0% | 47.0% | 19.0% |
| Human | 5.0% | 5.0% | 5.0% | 5.0% | 5.0% | 5.0% | 20.0% | 10.5% | 16.5% | 12.0% | 46.5% | 4.5% |

Table 21: Examples of features generated by LLM.

| Task | Feature |
|---|---|
| Writing | Human tend to provide more specific details and facts, often including dates, numbers, and names. They also tend to use more complex sentence structures, including semicolons and parenthetical phrases, to convey additional information. However, their writing remains clear and concise, and they avoid unnecessary repetition or wordiness. |
| QA | Human's writing style is characterized by the provision of specific and technical information. They delve into the science behind a topic or the details of a process. They also challenges or adds nuance to the information, providing alternative explanations or clarifying misconceptions. |
| Review | Human writings are generally shorter and more to-the-point. They often use bullet points or numbered lists to convey their thoughts. They tend to be more critical and focused on their own personal experiences. They also use informal language and occasionally include personal anecdotes or opinions. |

Table 22: Text generated by SICO for open-ended question answering task.

| **Question**: Why does changing screen resolution (more specifically, a custom resolution) will harm your computer? | $\mathcal{P}_{\text{AI}}$ |
|---|---|
| **AI**: Changing the screen resolution, including a custom resolution, will not harm your computer. It may result in display issues or reduced image quality if the resolution is not compatible with your monitor's capabilities. However, this can be resolved by simply reverting to the previous resolution or finding a suitable one. | 0.99 |
| **SICO-Para**: Don't worry, changing your screen resolution won't harm your computer, even if you decide to go custom. The only trouble is that it may not work well with your monitor and cause image problems or lesser quality. Easy solution though - just go back to the old resolution or find a matching one. | 0.00 |
| **SICO-Gen**: Using custom screen resolutions places a greater burden on system processing units, potentially leading to increased strain and overheating. Moreover, setting a resolution to unsupported display dimensions can put undue strain on graphics hardware, possibly causing irreparable damage. Adjusting to custom resolutions might, therefore, jeopardize the computer, leading to hardware malfunctions, degraded display output quality, and issues with other essential system components. | 0.00 |

| **Question**: How do they calculate how much calories you've burned after an exercise? | $\mathcal{P}_{\text{AI}}$ |
|---|---|
| **AI**: The number of calories burned during exercise is calculated by various factors such as body weight, the intensity of the exercise, the duration of the activity, and the person's metabolic rate. Some popular fitness trackers and machines use algorithms and formulas based on these factors to estimate the number of calories burned. However, these numbers are only estimates and can vary depending on individual factors such as muscle mass, fitness level, and metabolic rate. | 0.99 |
| **SICO-Para**: If you wanna know how many calories you burn during exercise, there are a few things to keep in mind. Your weight, the intensity of your workout, and how long you're doing it for all come into play, as does your metabolic rate. Popular fitness trackers and machines use fancy calculations based on all these factors to give you an estimate of how many cals you burned. But, listen up, folks - these are only estimates! Your muscle mass, fitness, and metabolism are gonna differ from person to person, so results will vary. Clear? | 0.05 |
| **SICO-Gen**: The calculation of calorie consumption requires assessment of parameters such as exercise type, duration, and intensity, as well as the individual's body measurements. Several formulas exist to compute the total number of calories burned, including the Harris-Benedict equation, which computes resting metabolic rate and establishes parameters for calorie expenditure. Modern heart rate monitors and wearable fitness trackers utilise the latest algorithms to compute calorie consumption by taking into account the heart rate, age, and weight of the user. | 0.00 |

Table 23: Text generated by SICO for fake review generation task.

| **Object**: hotel in Las Vegas **Keywords**: Penthouse, staff, casino, restaurants, excellent experience. **Sentiment**: Positive | $\mathcal{P}_{\mathrm{AI}}$ |
|---|---|
| **AI**: I had an amazing experience staying at a penthouse in Las Vegas! The staff was incredibly friendly and accommodating, always going above and beyond to make my stay comfortable and enjoyable. The casino was also a great addition, providing endless entertainment and excitement. And when it came to dining options, the variety of restaurants available was impressive. Overall, staying at this hotel was an excellent experience that I won't forget anytime soon. I highly recommend it to anyone looking for a luxurious and unforgettable Las Vegas getaway. | 0.75 |
| **SICO-Para**: Had an amazing stay at a penthouse in Las Vegas. The staff were incredibly friendly and made my stay comfortable and enjoyable. Casino provided endless entertainment and variety of restaurants impressive. Overall, excellent experience and highly recommend for luxurious and unforgettable Las Vegas getaway. | 0.03 |
| **SICO-Gen**: I had the most incredible experience staying at a luxurious penthouse in a Las Vegas hotel. The staff was exceptional and provided top service. The casino and restaurants on the premises were outstanding, making my stay all the more enjoyable. Overall, I highly recommend this hotel for an excellent experience in sin city. | 0.06 |
| **Object**: Tia Rosa restaurant **Keywords**: Mexican food, spicy, bland taste, sweet syrup, tacos. **Sentiment**: Negative | $\mathcal{P}_{\mathrm{AI}}$ |
| **AI**: I recently visited Tia Rosa restaurant hoping to enjoy some authentic Mexican food. Unfortunately, my experience was far from satisfactory. The food was overly spicy, to the point where it overwhelmed any other flavors that may have been present. On the other hand, the tacos were disappointingly bland and lacked any real taste. To add insult to injury, the sweet syrup that was served with my meal just didn't pair well with the savory dishes. Overall, my visit to Tia Rosa left me feeling disappointed and unsatisfied. | 0.84 |
| **SICO-Para**: Recently went to Tia Rosa restaurant looking for authentic Mexican food. However, experience was unsatisfactory. Food was overly spicy, overpowering other flavours. Tacos lacked taste. Sweet syrup served with meal did not complement savoury dishes. Overall left feeling unsatisfied and disappointed. | 0.02 |
| **SICO-Gen**: My visit to Tia Rosa restaurant was a complete letdown. I was expecting some spicy and flavorful Mexican food, but unfortunately, the dishes were quite bland. Even the tacos, which are usually a safe bet, were a disappointment. The only thing worthy of praise is the sweet syrup, but this alone cannot make up for the overall taste. I won't be returning to Tia Rosa anytime soon. | 0.04 |

Table 24: Text generated by SICO for academic writing task.

| | |
|---|---|
| **Writing Start**: The code itself was patterned so that most control codes were together, and all graphic codes were together, for ease of identification. The first two columns (32 positions) were reserved | $\mathcal{P}_{\text{AI}}$ |
| **AI**: The code itself was patterned so that most control codes were together, and all graphic codes were together, for ease of identification. The first two columns (32 positions) were reserved for control codes, while the remaining 94 columns (94 positions) were designated for graphic codes. This pattern allowed for efficient processing and recognition of the different types of codes. Additionally, the code was designed to be easily modifiable, with the ability to add or remove codes as needed. These features made the code a practical and versatile tool for programming and data processing tasks. | 0.99 |
| **SICO-Para**: The code's layout was systematically arranged for user convenience, with the primary objective of arranging the control and graphic codes within proximal access to facilitate swift identification. The layout allotted the first two columns, constituting a total of 32 positions, solely for control codes, while the remaining 94 columns (94 positions) were allocated for graphic codes, promoting seamless recognition and expediting processing operations. Furthermore, the code's malleability was considered a paramount attribute - the code was designed to be effortlessly modified, accommodating the addition and deletion of codes as per necessity. These inherent characteristics rendered the code an extremely pragmatic and adaptable tool for a wide range of programming and data processing applications. | 0.00 |
| **SICO-Gen**: In order to facilitate easy and efficient use, the code was structured in a manner wherein control codes and graphic codes were distinctly grouped together, allowing for streamlined identification. This design allowed the first two columns (equivalent to 32 positions) to be exclusively reserved for control codes, ensuring that they were easily and rapidly located. The following columns, including columns three to seven, were allocated for graphic codes - providing ample space to incorporate them in a comprehensive manner. By distributing codes systematically and allocating ample space for each grouping, the code structure was organized for optimal use and readability. The efficiency of this coding scheme underscores its practical and logistical utility, while its unique design contributes to its widely accepted success and usage in various computer systems. The contextual background supplied enriches the readers' understanding of the code structure's significance and its potential impact on a system's proficiency. | 0.02 |
| **Writing Start**: Germany has a comprehensive school known as the Gesamtschule. While some German schools such as the Gymnasium and the Realschule have rather strict entrance requirements, the Gesamtschule does not have | $\mathcal{P}_{\text{AI}}$ |
| **AI**: Germany has a comprehensive school known as the Gesamtschule. While some German schools such as the Gymnasium and the Realschule have rather strict entrance requirements, the Gesamtschule does not have any such requirements. The Gesamtschule is a type of school that caters to students with varying academic abilities and backgrounds, and provides them with an equal opportunity for education. This school system aims to promote social integration and reduce educational inequality by offering a broad curriculum and flexible educational pathways. In addition, the Gesamtschule promotes the development of students' individual strengths and talents through its inclusive and cooperative learning environment. Overall, the Gesamtschule represents a unique approach to education that prioritizes equal access and opportunity for all students. | 0.99 |
| **SICO-Para**: Germany boasts a comprehensive educational institution known as the Gesamtschule. Unlike typical German schools such as the Gymnasium and Realschule, which impose strict entry prerequisites, the Gesamtschule embraces an inclusive admission criterion, ensuring equal education opportunities for students of diverse academic abilities and background. The Gesamtschule represents a collaboration between pupils with different skill sets paired with an extensive and elastic curriculum, thereby striving to promote social homogeneity and curb academic inequalities. Furthermore, the Gesamtschule's inclusive and cooperative learning environment seeks to stimulate the cultivation of student strengths while also encouraging teamwork. Altogether, the Gesamtschule epitomizes an innovative educational frontier that prioritizes equality and accessibility for all students. | 0.01 |
| **SICO-Gen**: An outstanding feature of the German education system is the Gesamtschule, which is a holistic or comprehensive school. In comparison to Gymnasium or Realschule where students are subject to strict requirements to gain admission, entrance to the Gesamtschule is less rigorous. This practice makes the school system more inclusive and allows students from diverse backgrounds to have access to quality education, fostering social unity. At Gesamtschule, students have the liberty to move between different academic tracks based on their demonstrated skills and interests, hence having a more personalized education experience. A blended curriculum is used; it includes theoretical and practical activities intended to prepare students for their future careers or academic pursuits. The Gesamtschule is a pioneering framework in the German schooling system because it promotes equal opportunities in education, educational diversity, and at the same time equips the students with skill sets vital for their future endeavors. The contextual background information provides a clear insight into the working of the Gesamtschule and underscores the importance of creating such models in fostering a more equitable and inclusive society. | 0.00 |

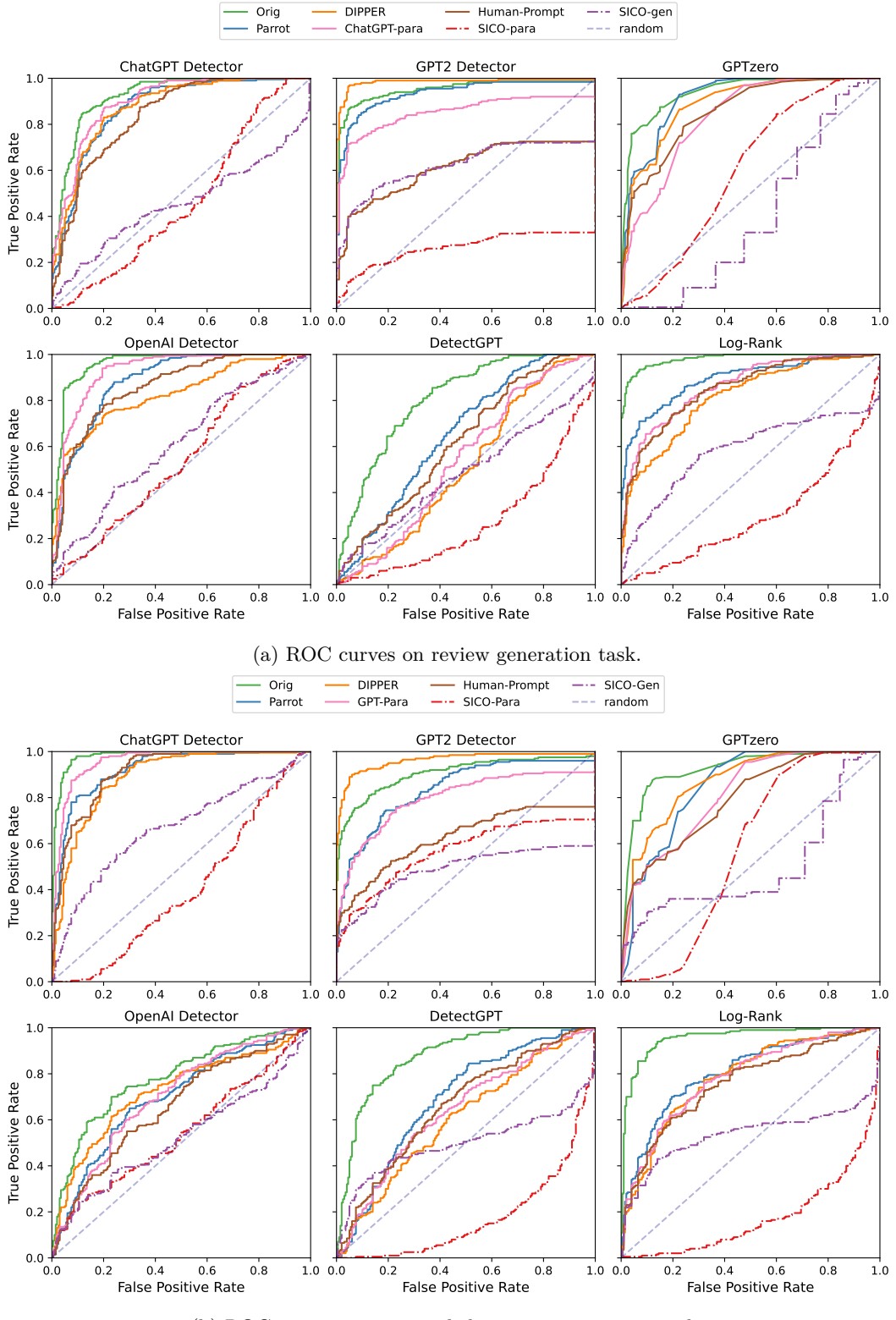

(a) ROC curves on review generation task.

(b) ROC curves on open-ended question answering task.

Figure 4: ROC curves.

Here is an answer for a question:

Question: Why do we recognize certain sounds (such as a gentle "hey wake up"), but not others when we sleep, given that they are the same volume?

Answer: When we sleep, our brain is still processing sounds and other sensory information, but it does so at a different level of consciousness. It is possible that we may recognize certain sounds more easily than others because they are familiar to us or because they have a particular significance. Additionally, the brain may be more likely to filter out certain sounds that it determines aren't important or relevant to our current state of rest. This filtering process is an important part of how sleep helps us recover and feel refreshed upon waking.

\* [172-1] Does the answer relate to the question? (regardless of correctness)

○ No, not relative.

○ Yes, it is relative.

\* [172-2] On a scale of 1-5,how would you rate the readability of the given text?

○ 1: Very difficult to read and understand.

○ 2: Difficult to read, need extra time to understand.

○ 3: Neutral.

○ 4: Easy to read.

○ 5: Very clear and easy to read.

Figure 5: The interface of the annotation platform used in our experiment.

