# OpenReview forum: "Large Language Models can be Guided to Evade AI-generated Text Detection"
_TMLR — Accepted by TMLR_

### Review · Reviewer_m6ge · 2024-02-02

**Summary Of Contributions:**

The authors train LLMs to avoid being detected as AI text generators through a technique they call Substitution-based In-Context example Optimization (SICO) as a fancy form of prompt engineering. The core idea is to systematically perturbs the exemplars within a prompt to produce diversity and variation in its output. This substitution procedure is guided by a "proxy detector", acting similar to a discriminator in a GAN.  They then test on three tasks: (a) essay writing (b) open domain QA and (c) review generation. Compared to paraphrasing baselines, SICO decreases AUC by 0.5 on average, while still maintaining high naturalness.  This is validated through human evaluation and experimentation on Reddit.

**Audience:**

Yes

**Broader Impact Concerns:**

The authors mention that their "aim is to raise awareness within the broader community about the vulnerabilities of existing AI-generated text detection systems", but many people are already raising the alarm on the danger of AI. There are entire "Safety Departments" concerned about alignment at all major AI research labs. Without an improved method for detecting AI-text (such as improved watermarking), the work seems to only make the problem worse.

**Claims And Evidence:**

Yes

**Requested Changes:**

The paper would be greatly improved if a novel method was introduced that was able to out-smart SICO and all the other paraphrase baselines.  As things currently stand, the ideas introduced only make the situation worse for AI dishonesty.
Also run across many more variations and trials to show that any technique is stable, which is often not the case for prompt based methods.

**Strengths And Weaknesses:**

Strengths:
  - New idea for prompting where SICO systematically perturbs the exemplars within a prompt to produce diversity and variation in its output.
  - Apply an LLM to write natural language descriptions of what makes text human-like vs machine-like.  Then uses these features to construct a better exemplars for the downstream LLM.
  - A proxy is employed during "training" to iteratively find better examples for the Constructed Prompt.
  - Showcases the strong ability of LLMs to perform well in few shot learning scenarios (~40 examples) and careful prompt engineering.

Weaknesses:
  - The SICO technique is inherently based on prompting and thus is highly unstable due to the brittle nature of prompt engineering.  The authors run their experiment with one run, and fail to show that the method generalizes across multiple trials.
  - Detecting AI generated text to avoid spam or misinformation is a worthwhile task, and despite talking about the problem extensively, the introduced method does *not* address the issue.  Instead, they actually make the problem worse by teaching others how to avoid the existing detection methods.
  - Requires access to a proxy detector and then greedily makes substitutions.  This ends up as a brute force method rather than anything particularly insightful.
  - Requires many examples of human demonstrations to work. This defeats the whole purpose of the task because suppose I want to write fake product reviews.  If I have already written so many for demonstration, then my task of "tricking people to buy my product" is already accomplished.
  - Only tested with GPT 3.5 and Vicuna rather than GPT 4 or GPT 4 Turbo, or Claude, or Mixtral, or many of the other open source options. This also calls into question the generality of results.

---

> ### Author Response · Authors · 2024-03-26
> **Response to Reviewer m6ge Part 1**
>
> ## Weakness
>
> **[w1] The SICO technique is inherently based on prompting and thus is highly unstable due to the brittle nature of prompt engineering. The authors run their experiment with one run, and fail to show that the method generalizes across multiple trials.**
>
> We fully agree with your assertion that prompt engineering, particularly when it involves human labor, can be brittle. However, the whole process of SICO is stable. To address your concern, we conducted experiments to confirm the stability of our prompt optimization process.  The results are presented in Appendix B.2.
>
> Here are the techniques we employed in SICO to mitigate the instability of the prompt optimization process:
>
> - During the feature extraction step, we sample multiple feature texts and use prompt evaluation (Eq.1) to select the best one. Details can be found in Appendix A.1.
> - In-context examples are consistently optimized in the same direction $N$ times, reducing instability throughout the process.
>
> **[w2] Detecting AI generated text to avoid spam or misinformation is a worthwhile task, and despite talking about the problem extensively, the introduced method does not address the issue. Instead, they actually make the problem worse by teaching others how to avoid the existing detection methods.**
>
> Thanks for your insightful reiview. We fully agree that the detection of AI-generated spam or misinformation is worthwhile. However, this work focuses on detecting AI-generated text, rather than focusing on particular tasks such as spam detection. Consequently, we follow the typical task settings in this domain [1-4], which primarily include QA and writing completion tasks. We believe the three tasks presented in our paper accurately represent our research topic and can be generalized to other tasks.
>
> For the safety concern, we proposed an ensemble method against SICO in Section 6. The results show that this approach significantly improves detection capabilities against SICO text, while maintaining the ability to identify the original AI-generated text.
>
> **[w3] Requires access to a proxy detector and then greedily makes substitutions. This ends up as a brute force method rather than anything particularly insightful.**
>
> Yes, the greedy substitution strategy is simple and straightforward. However, it's critical to notice that the entire framework is insightful, which aligns the LLM's output with specific criteria by adjusting in-context examples to meet a proxy criterion. This framework provides a new direction for in-context learning and benefits the community.
>
> **[w4] Requires many examples of human demonstrations to work. This defeats the whole purpose of the task because suppose I want to write fake product reviews. If I have already written so many for demonstration, then my task of "tricking people to buy my product" is already accomplished.**
>
> Yes, if you're looking to write a few (less than 40) product reviews, SICO will not help. However, for a malicious organization that aims to write hundreds of reviews for various products, it only requires multiple LLM inferences after constructing one SICO prompt. SICO helps in the case of massive AI text generation.
>
> **[w5] Only tested with GPT 3.5 and Vicuna rather than GPT 4 or GPT 4 Turbo, or Claude, or Mixtral, or many of the other open source options. This also calls into question the generality of results.**
>
> We conducted experiments with additional LLMs, including GPT-4 and WizardLM. The results indicate that SICO with different LLMs maintains a high degree of detection evasion performance. Please refer to the results in Appendix B.4. We hope these new results address your concerns.
>
> ### References
>
> [1] Kalpesh Krishna, Yixiao Song, Marzena Karpinska, John Wieting, Mohit Iyyer: Paraphrasing evades detectors of AI-generated text, but retrieval is an effective defense. NIPS 2023
>
> [2] Eric Mitchell, Yoonho Lee, Alexander Khazatsky, Christopher D. Manning, Chelsea Finn: DetectGPT: Zero-Shot Machine-Generated Text Detection using Probability Curvature. ICML 2023
>
> [3] Vinu Sankar Sadasivan, Aounon Kumar, Sriram Balasubramanian, Wenxiao Wang, Soheil Feizi: Can AI-Generated Text be Reliably Detected? arxiv 2023
>
> [4] Xinlei He, Xinyue Shen, Zeyuan Chen, Michael Backes, Yang Zhang: MGTBench: Benchmarking Machine-Generated Text Detection. arxiv 2023

---

> > ### Author Response · Authors · 2024-03-26
> > **Response to Reviewer m6ge Part 2**
> >
> > ## Requested Changes
> >
> > **[R1] The paper would be greatly improved if a novel method was introduced that was able to out-smart SICO and all the other paraphrase baselines.**
> >
> > We proposed an ensemble method to defend against SICO text, which utilizes two detectors trained separately to identify original AI-generated text and SICO text. The results show that this approach significantly improves detection capabilities against SICO text, while maintaining the ability to identify the original AI-generated text. We hope these new results address your concerns.
> >
> > Besides, open-sourcing the attack technique also benefits the community by revealing vulnerabilities in detectors, thereby providing a direction to enhance them. If some malicious individuals discover the same safety issue but keep it secret, the consequences could be much more severe.
> >
> > **[R2] Run across many more variations and trials to show that any technique is stable**
> >
> > We examined the stability of SICO by running multiple trials. The results show that SICO is able to consistently construct effective prompts for detection evasion. Please refer to Section Appendix B.2. We hope these results could address your concerns.

---

### Review · Reviewer_XdS7 · 2024-03-06

**Summary Of Contributions:**

The paper presents a paraphrase-based method for evading generative text detectors called SICO. The proposed methods requires a small training dataset (40 human-written examples). SICO is evaluated on 3 tasks and against 6 current detection models/methods. The method reports good performance in fooling current text detection methods. The authors also evaluate the quality of the modified text to assess the impact of SICO on human preferences, where they find readability + task completion rate performs competitively against human-written text and outperforms

**Audience:**

Yes

**Broader Impact Concerns:**

- Deploying the AI text to reddit to collect human feedback on SICO-generated responses seems a bit fraught ethically. A cleaner, less problematic experiment here would be been do a more controlled experiment, e.g., presenting message threads on mechanical turk and having annotators vote/score responses. Technically can a bot even post to Reddit without direct solicitation (see https://www.reddit.com/wiki/bottiquette/) Some discussion of this would be appreciated.

**Claims And Evidence:**

Yes

**Requested Changes:**

The following points could be elaborated on and would improve my scoring of the manuscript

- The other methods seem to be trained models or simple prompts. SICO provides a method to iteratively improvement of the prompt.

- Any reason you don't include the watermarking result in the main text vs. the suppliment? SICO seems to help here as well, though not as strongly as the other evasion categories

- Some discussion on the "arms race" aspect of detection models. How long do we expect this method to continue working? It seems like having a method that relies on some known asset like WordNet is something that future detectors can easily incorporate into training future detectors.

- Misc Clarifications:
"We restrict substitutions to content words that carry meanings and ensure that the substitution would not change the part-of-speech tags." What is meant by content words that carry meanings? Is this something specific to WordNet, noun phrases, etc?

**Strengths And Weaknesses:**

Strengths:

- Comprehensive experiments against many baselines and realistic tasks.
- Nice ablations of components of SICO (Table 4)
- Method illustrates how futile generation detection methods are.

Weaknesses:

- Why is human evaluation on compared against DIPPER vs other methods?
- Word-level substitution via WordNet seems less powerful than letting an LLM do it's own substitution?

---

> ### Author Response · Authors · 2024-03-26
> **Response to Reviewer XdS7**
>
> We thank you for the quality and precision of your feedback. We respond to your points below in the "weakness" and "requested changes" section.
>
> ## Weaknesses:
>
> **[w1] Why is human evaluation on compared against DIPPER vs other methods?**
>
> Thanks for your thoughtful question. The reason for selecting DIPPER is that its evasion performance is better than other baselines on average. And DIPPER is specifically trained to evade AI detectors. Considering the high cost of hiring human annotators, we only select DIPPER as a representative paraphraser baseline in the human evaluation experiment.
>
> **[w2] Word-level substitution via WordNet seems less powerful than letting an LLM do it's own substitution?**
>
> Yes, intuitively, LLM-generated substitution is better than WordNet in terms of diversity and fluency. However, prompting the LLM to generate substitutions for each word requires extra computational cost and time compared with the WordNet approach. So, in SICO, we adopted WordNet to limit costs and increase usability.
>
> ## Requested changes
>
> **[R1] The other methods seem to be trained models or simple prompts. SICO provides a method to iteratively improvement of the prompt.**
>
> Thanks for your advice. We elaborated the key difference in Section 2.1.
>
> **[R2] Any reason you don't include the watermarking result in the main text vs. the suppliment? SICO seems to help here as well, though not as strongly as the other evasion categories**
>
> The key reason is that SICO is not specially designed for watermarking mechanisms. We attribute the evasion performance against watermark more to the powerful capabilities of the LLM itself, rather than the contribution of SICO. We add this discussion in Appendix G.
>
> **[R3] Some discussion on the "arms race" aspect of detection models. How long do we expect this method to continue working? It seems like having a method that relies on some known asset like WordNet is something that future detectors can easily incorporate into training future detectors.**
>
> Thank you for your insightful perspective. We add discussion in Section 6.2. Here is a brief discussion:
>
> We expect the new attack method to continue working until the defender collects enough adversarial examples and trains a new detector. Then, the attacker may invent a new technique to evade the current detector. This "arms race" is based on the assumption that both sides are willing to share information like attack techniques or detector API. If either side stops sharing, the race will stop.
> In practice, the defender has a superior position in this race, as it can restrict access to the detector and prevent attackers from evolving methods. Besides, the defender, typically the big company, has more resources like money and computing power.
>
> **[R4] Misc Clarifications: […] What is meant by content words that carry meanings? Is this something specific to WordNet, noun phrases, etc?**
>
> We add an extra explanation in Section 3.2. Here is a brief explanation:
>
> We define content words as single words that are nouns, verbs, adjectives, and adverbs. We replace single words, not phrases. In our experiment, we utilize the Stanford POS Tagger [1] to determine the POS of words.
>
> [1] Kristina Toutanova et al. Feature-rich part-of-speech tagging with a cyclic dependency network. NAACL 2003
>
> ## Safety Concerns
>
> To to minimize the social impact, we limit the number of responses to 40 and deleted
> them after collecting results. We only post reviews in the Reddit community that not banned ChatGPT.
> Addtionally, we provide a defensive method against SICO. Experiments demonstrate its effectiveness.

---

> ### Author Response · Authors · 2024-04-22
> **Response to Ethic Concerns**
>
> We agree with reviewer `8CXt` that the follower may do the same thing on social platform, potentially leading to harmful results. Therefore, we removed all relevant information about the real-life experiments from the paper. Please review the latest manuscript.
>
> We are more than happy to answer any additional questions. Your feedback will be greatly appreciated.

---

### Review · Reviewer_8CXt · 2024-03-14

**Summary Of Contributions:**

Finetuned classifiers and statistical methods are conventional methods used to distinguish whether the response is generated by humans or AI. This paper focuses on evaluating the vulnerability of these detectors. In this paper, the authors propose a novel Substitution-based In-Context example Optimization method (SICO) to automatically construct prompts for evading the detectors. Specifically, SICO iteratively rephrases the in-context examples to offer representative demonstrations for LLMs to generate undetectable text. Notably, the rephrasing procedure is directed by a proxy detector.

The main contributions of this paper are:
1. Compared to conventional methods that apply external paraphrase, SICO takes advantage of LLM, and constructs prompts to evade detectors.
2. The authors assess SICO across various tasks, including academic essay writing, question answering and fake review generation. The results demonstrate the strength of SICO, leading to a significant decrease in AUC for existing detectors, revealing the vulnerability of current methods.
3. Lastly, human evaluation in the wild is also included to validate the effectiveness of SICO.

**Audience:**

Yes

**Broader Impact Concerns:**

The authors have well explained the impact in the paper, especially the negative impact of the human evaluation experiment.

There is one ethical issue in Section 4.5 about the real-life experiments. I personally encourage removing this part, because:
1. It doesn't add much evidence to the paper, the number of "Dislike", "Like" or "Comment" doesn't necessarily show whether the text is generated more like a human's. It might only be about the usefulness.
2. This evaluation is harmful to the community, even though all AI-generated responses on Reddit have been deleted. If the followers do similar evaluation in the future, it might hurt the platform.

**Claims And Evidence:**

Yes

**Requested Changes:**

Overall, I recommend an acceptance of this paper, since most claims are well supported. However, there are still two more requested changes:
1. Rewriting of the method section, including more examples/figures to highlight the key factors;
2. A better human evaluation setting or the author can convince me that the current setting is desired. This is the key factor for the acceptance.

**Strengths And Weaknesses:**

Strengths:
1. The proposed method is well-motivated. The authors try their best to give an explanation for each design.
2. Most of the experimental settings are convincing. And the results of SICO are promising.
3. The ablation study is thorough.


Weaknesses:
1. The method section, i.e. the 3rd section, is not easy to follow. It's better to give more detailed examples for each design and highlight the key factors.
2. The human evaluation in the wild is not convincing. Response with high likes and discussions could only demonstrate the usefulness of the response instead of showing whether the response is generated by humans or AI.

---

> ### Author Response · Authors · 2024-03-26
> **Response to Reviewer 8CXt**
>
> Thanks for your instructive review. We respond to your points below in the "weakness" and "requested changes" section.
>
> **[W1&R1]Rewriting of the method section, including more examples/figures to highlight the key factors**
>
> We have modified Section 3 in the revised manuscript to clarify and highlight the key factors.
>
> **[W2&R2] Response with high likes and discussions [..] (not) showing whether the response is generated by humans or AI. A better human evaluation setting or the author can convince me that the current setting is desired.**
>
> We agree that the current experiment does not show the imperceptibility of SICO text. So we added a human evaluation experiment of imperceptibility in Section 4.4. The results demonstrate that SICO reduces the probability of being recognized by human annotators. We hope these new results address your concerns.

---

> ### Author Response · Authors · 2024-04-22
> **Response to Ethic Concerns**
>
> Thanks for your timely and instructful feedback! We agree that the follower may do the same thing, potentially leading to harmful results. Therefore, we removed all relevant information about the real-life experiments from the paper. Please review the latest manuscript.

---

### Author Response · Authors · 2024-03-26
**Common response to all reviewers**

We thank all reviewers for their comments. We have uploaded a revised version of the paper that we hope represents a meaningful improvement and addresses the raised concerns, in conjunction with our responses here. We have specified in the reviewer-specific responses where any relevant revisions have been made. All changes in the paper are highlighted in (dark-ish) red.

---

### Decision · Action_Editor_yTUt · 2024-04-26

**Recommendation:** Accept as is

**Comment:**

The paper has improved following the reviewer suggestions, including additional experiments that strengthen the work. An earlier version included an experiment that did not bring much to the paper but also raised ethical concerns; that section has been removed.

**Audience:**

The topic is timely and of interest to many in the TMLR audience, as the reviewers agree.

**Claims And Evidence:**

The reviewers agree that the paper presents a useful method and backs its main claims adequately.